# Rosmanol and Carnosol Synergistically Alleviate Rheumatoid Arthritis through Inhibiting TLR4/NF-κB/MAPK Pathway

**DOI:** 10.3390/molecules27010078

**Published:** 2021-12-23

**Authors:** Lianchun Li, Zhenghong Pan, Desheng Ning, Yuxia Fu

**Affiliations:** Guangxi Key Laboratory of Functional Phytochemicals Research and Utilization, Guangxi Institute of Botany, Chinese Academy of Sciences, Guilin 541006, China; llc@gxib.cn (L.L.); ndsh@gxib.cn (D.N.); fyx@gxib.cn (Y.F.)

**Keywords:** rheumatoid arthritis, rosmanol, carnosol, *Callicarpa longissima*, TLR4/NF-κB/MAPK, synergistic effect

## Abstract

*Callicarpa**longissima* has been used as a Yao folk medicine to treat arthritis for years in China, although its active anti-arthritic moieties have not been clarified so far. In this study, two natural phenolic diterpenoids with anti-rheumatoid arthritis (RA) effects, rosmanol and carnosol, isolated from the medicinal plant were reported on for the first time. In type II collagen-induced arthritis DBA/1 mice, both rosmanol (40 mg/kg/d) and carnosol (40 mg/kg/d) alone alleviated the RA symptoms, such as swelling, redness, and synovitis; decreased the arthritis index score; and downregulated the serum pro-inflammatory cytokine levels of interleukin 6 (IL-6), monocyte chemotactic protein 1 (MCP-1), and tumor necrosis factor α (TNF-α). Additionally, they blocked the activation of the Toll-like receptor 4 (TLR4)/nuclear factor κB (NF-κB)/c-Jun N-terminal kinase (JNK) and p38 mitogen-activated protein kinase (MAPK) pathways. Of particular interest was that when they were used in combination (20 mg/kg/d each), the anti-RA effect and inhibitory activity on the TLR4/NF-κB/MAPK pathway were significantly enhanced. The results demonstrated that rosmanol and carnosol synergistically alleviated RA by inhibiting inflammation through regulating the TLR4/NF-κB/MAPK pathway, meaning they have the potential to be developed into novel, safe natural combinations for the treatment of RA.

## 1. Introduction

Rheumatoid arthritis (RA) is a chronic autoimmune disease that is characterized by the presence of autoantibodies, lasting synovitis, and systemic inflammation. It can cause joint destruction and disability with a prevalence of 0.5–1%, leading to considerable costs to both the individual and the community [1]. Nonsteroidal anti-inflammation drugs, glucocorticoids, disease-modifying anti-rheumatic drugs, and biological agents are the currently used therapeutic drugs for RA. However, they have different side effects, including gastrointestinal, liver, pulmonary, hematological, neurological, and renal toxicities, as well as risks of serious infections [2]. Therefore, it is urgently needed to develop novel alternative drugs with less adverse effects for the treatment of RA. To meet this requirement, more anti-inflammatory medicinal plants traditionally used to treat RA have been considered, and many medicinal plant-derived anti-RA drugs have been successfully developed, including glucosides of *Tripterygium wilfordii* and total glucosides of paeony and sinomenine [3,4,5]. These successful stories demonstrate that undertaking an ethnopharmacology-based investigation is an effective strategy for the development of anti-RA drugs.

*Callicarpa longissima* (Hemsl.) Merr. is a shrub that widely grows in southern China, which has been used as a Yao folk medicine to treat arthritis, common cold, cough, bleeding, and abdominal pain [6]. The reported phytochemical composition of *C. longissima* include glycosides [6], terpenoids [7,8], phenylpropanoids, lignans, steroids, essential oils, and flavonoids [9]. Some of these compounds exhibit considerable physiological functions, such as anti-cancer [7], skin-whitening [8], anti-inflammatory [7,9], and antioxidant effects [10]. Unfortunately, no study on the anti-arthritic effect of this ethnomedicine has been reported yet.

Recently, we investigated the phytochemicals of *C. longissima* and isolated two phenolic diterpenoids, rosmanol and carnosol. Rosmanol and carnosol are abundant in this medicinal plant, with contents of no less than 0.46 mg/g and 2.37 mg/g (Appendix A), respectively, and could be regarded as its characteristic constituents. Interestingly, they are also present in *Rosmarinus officinalis* (rosemary), a medicinal and edible plant native to the Mediterranean region [11]. As major constituents of rosemary extracts, they are commonly used as flavoring agents in cooking and as antioxidants in food preservation, which have been approved by the European Union [12]. Additionally, rosmanol and carnosol in the dose range of 50–200 mg/kg/d did not exhibit any signs of acute toxicity in male Swiss mice [13,14]. Therefore, they are safe for human consumption.

The anti-inflammatory effects of rosmanol and carnosol have been investigated mainly in vitro with cell experiments. A previous study reported that rosmanol potently decreased the expression of inducible nitric oxide synthase (iNOS) and cyclooxygenase 2 in lipopolysaccharide-stimulated RAW 264.7 cells, which were mediated by inhibiting the activation of nuclear factor κB (NF-κB), signal transducer and activator of transcription-3, CCAAT/enhancer binding protein (C/EBP), and mitogen-activated protein kinase (MAPK) signaling pathways [15]. Chang et al. found that rosmanol inhibited not only the migration and proliferation of rat-fibroblast-like synoviocytes, but also the endothelial tube formation of human umbilical vein endothelial cells, which were mediated via regulating the C/EBP δ signaling pathway [16]. Schwager et al. disclosed that carnosol strongly inhibited the production of nitric oxide and prostaglandin E_2_ and significantly downregulated the gene expression of iNOS, cytokines, and chemokines in both macrophages and chondrocytes, primarily by regulating the NF-κB signaling pathway [17]. Using a carrageen-induced paw edema mouse model, we found that the combination of rosmanol and carnosol exhibited more powerful anti-inflammatory effects than either compound alone (Appendix A), indicating that they synergistically suppressed inflammation.

Toll-like receptor 4 (TLR4) is a member of the Toll-like receptor family. Its downstream signaling pathways includes NF-κB and MAPKs [18]. The NF-κB is an important regulator of pro-inflammatory genes expression, such as tumor necrosis factor α (TNF-α), interleukin 6 (IL-6), IL-1β, and cyclooxygenase 2 (Cox-2), and plays a key role in the regulation of inflammation [19]. MAPK family proteins, including the extracellular-signal-regulated kinases, c-Jun N-terminal kinase (JNK), and p38 in mammals, are tightly associated with RA pathogenesis [20]. JNK regulates the expression of matrix metalloproteinases (MMPs), as well as the proliferation, migration, and invasion of synoviocytes and the destruction of joints [21,22]. P38 is critical for RA pathogenesis, as its activation involves almost all aspects of RA-related pathologies, including the expression of pro-inflammatory cytokines, synovitis, cartilage degradation, bone destruction, and angiogenesis [23,24]. Therefore, the TLR4/NF-κB/MAPK pathway is an important target for the treatment of RA.

Altogether, based on the reported studies, it is reasonable to speculate that rosmanol and carnosol are anti-arthritic constituents of *C. longissima*. Inspired by their synergistic effect in carrageen-induced paw edema mice, we assume that rosmanol and carnosol may synergistically alleviate RA. To confirm our hypothesis, the pharmacodynamics and effects on TLR4/NF-κB/MAPK pathway of rosmanol, carnosol, and the combination of both were investigated in this study with a collagen-induced arthritis (CIA) DBA/1 mouse model.

## 2. Materials and Methods

### 2.1. Plant Materials

The branches and leaves of *C. longissima* were collected from the Botanic Garden of Guangxi Institute of Botany, Chinese Academy of Sciences (CAS) in October 2019, and were identified by Associate Professor Yu-song Huang (Guangxi Institute of Botany, CAS). A voucher specimen (CTM201916) was deposited at Guangxi Key Laboratory of Functional Phytochemicals Research and Utilization, Guangxi Institute of Botany, CAS.

### 2.2. Preparation of Rosmanol and Carnosol

The dried branches and leaves of *C. longissima* (5.0 kg) were extracted with 95% ethanol at room temperature three times. The ethanol extract was concentrated under reduced pressure at 50 °C to give a crude extract (557 g), which was dissolved in water and partitioned successively with petroleum ether (PE) and ethyl acetate. The ethyl acetate extract (251.3 g) was subjected to a 100–200 mesh silica gel column (Qingdao Marine Chemical Factory, Qingdao, China) and eluted with PE-Me_2_CO (from 20:1 to 1:1, *v*/*v*) to give five fractions (Fr.1–Fr.5). Fr.5 (18.3 g) was performed on a silica gel column (PE-Me_2_CO, from 20:1 to 2:1, *v*/*v*) and subsequently purified on semi-preparative HPLC column (Agilent Technologies Inc., Santa Clara, CA, USA) with CH_3_CN/H_2_O (50:50) to give **1** (421.2 mg, purity >95% (Appendix A)). Fr.2 (36.2 g) was purified repeatedly on a silica gel column (PE-Me_2_CO, 5:1, *v*/*v*) to yield **2** (2.1 g, purity > 95% (Appendix A)). HR-ESI-MS data was measured on an LC/MS-IT-TOF mass spectrometer (Shimadzu Co., Ltd., Kyoto, Japan) and NMR spectra were recorded with an AVANCE III HD 500 spectrometer at 25 °C (Bruker Co., Ltd., Ettlingen, Germany). Compounds **1** and **2** were identified as rosmanol and carnosol (Figure 1), respectively, by direct comparison of their MS and NMR spectral data (Appendix A) with the literature [25].

### 2.3. Chemicals and Antibodies

Bovine type II collagen (BIIC, 20022), Complete Freund’s Adjuvant (CFA, 7009), and Incomplete Freund’s Adjuvant (IFA, 7002) were acquired from Chondrex (Redmond, WA, USA). IL-6 (E-EL-M0044c), monocyte chemotactic protein 1 (MCP-1, E-EL-M3001), and TNF-α (E-EL-M0049c) enzyme-linked immunosorbent assay (ELISA) kits were obtained from Elabscience Biotechnology (Wuhan, China). The total RNA extraction kit (CW0560S) and cDNA synthesis kit (CW0744M) were obtained from CoWin Biosciences (Beijing, China). The quantitative real-time polymerase chain reaction (PCR) kit (FP205) was acquired from Tiangen Biotech (Beijing, China). RIPA lysis buffer (P0013B), phenylmethylsulphonyl fluoride (PMSF, ST506), sodium orthovanadate (S1873), the BCA protein quantification kit (P0012S), enhanced chemiluminescence reagents (ECL, P0018S), horse radish peroxidase (HRP)-labeled goat anti-mouse (A0286), and anti-rabbit (A0277) secondary antibodies were acquired from Beyotime (Nantong, China). Primary antibodies against glyceraldehyde-3-phosphate dehydrogenase (GAPDH, 60004-1-Ig) and TLR4 (66350-1-Ig) were obtained from Proteintech (Wuhan, China). Primary antibodies against MyD88 (4283S), NF-κB p65 (8242T), p-p65 (3033T), JNK (9252T), p-JNK (4668T), p38 (8690T), and p-p38 (4511T) were purchased from Cell Signaling Technology (Boston, MA, USA).

### 2.4. Animals

Thirty male DBA/1 mice aged 5−6 weeks were obtained from Changzhou Cavens Laboratory Animals Limited Company (Changzhou, China, certification No. SCXK (Su) 2016−0010). The mice were given free access to food and water. All animal care procedures followed the Guidelines for the Care and Use of Laboratory Animals from the Ministry of Science and Technology of China. The animal experimental protocols used in this research were approved by the Laboratory Animal Management and Ethics Committee of Guangxi Institute of Botany, CAS (Guilin, China).

### 2.5. Establishment of CIA DBA/1 Mouse Model

After 7 days of adaption to laboratory conditions, all mice were randomly divided into 5 groups (6 mice/group), namely a drug-untreated control group (normal), BIIC-immunized group (CIA model), BIIC combined with rosmanol-treated (40 mg/kg) group, BIIC combined with carnosol-treated (40 mg/kg) group, and BIIC combined with the combination-treated (20 mg/kg rosmanol and 20 mg/kg carnosol) group. The protocol used for establishment of CIA DBA/1 mice was described by Miyoshi and Liu [26]. Briefly, on day 0, 100 µL of emulsion of BIIC and CFA (containing 100 μg BIIC) was subcutaneously injected into the base of each mouse tail, except for the normal group. A booster injection of the emulsion of BIIC and IFA was performed 21 days after the initial immunization. After the booster immunization, rosmanol, carnosol, or their combination was administered by intragastric gavage once daily on days 21–42, respectively, whereas the normal and model group mice were fed with normal saline instead (Figure 2a).

### 2.6. Evaluation of Arthritis Severity

The severity of arthritis was evaluated on days 28, 35, and 42 using an established macroscopic scoring system of 0–4 per paw as described previously [27]. The severity of arthritis in each paw was scored as follows: 0 = normal joint; 1 = swelling of one joint (toe/wrist/ankle/footpath); 2 = swelling of more than one joint; 3 = swelling of all joints; 4 = bursting of the skin, dysfunction, or distortion of the joint. The cumulative score for all four paws in each mouse was used as an arthritis index to represent the overall disease severity and progression.

### 2.7. Hematoxylin and Eosin (H&E) Staining

Paraffin sections of synovium tissues were stained with H&E. The right hind limb specimens were fixed with 10% (*v*/*v*) neutral formalin for 24 h, then embedded in paraffin and sliced into 4-μm-thick tissue sections. H&E staining was performed according to protocols described previously [28].

### 2.8. ELISA

On day 42, all mice were euthanized by CO_2_ asphyxiation. Blood samples from each mouse were collected immediately, coagulated naturally at room temperature for 1 h, and then centrifuged by 3000× *g* for 10 min at 4 °C. The IL-6, MCP-1, and TNF-α levels of the supernatant (serum) were determined with ELISA kits according to the manufacturer’s instructions. The absorbance at the wavelength of 450 nm was determined on a SPARK microplate reader (Tecan, Männedorf, Switzerland).

### 2.9. Quantitative Real-Time PCR

Total RNA was prepared from mice synovial tissues and used as a template for first strand cDNA synthesis. Quantitative real-time PCR analysis was performed with the QuantStudio 6 System (ABI, Foster, CA, USA). Primer sequences used in real-time PCR were as follows: GAPDH forward and reverse primers, 5′-ATGGGTGTGAACCACGAGA-3′ and 5′-CAGGGATGATGTTCTGGGCA-3′; TLR4 forward and reverse primers, 5′-GCCCTACCAAGTCTCAGCTA-3′ and 5′-CTGCAGCTCTTCTAGACCCA-3′; MyD88 forward and reverse primers, 5′-CCCACTCGCAGTTTGTTG-3′ and 5′- CACCTGTAAAGGCTTCTCG-3′; p65 forward and reverse primers, 5′-CACCGGATTGAAGAGAAGCG-3′ and 5′- AAGTTGATGGTGCTGAGGGA-3′. The reaction mixtures contained 10 μL of SYBR Green Master Mix, 0.4 μL of ROX Reference Dye (50×), 1 μL of cDNA, 0.4 μL of forward and reverse primers (10 μM), and 7.8 μL of RNase-free water. The amplification protocols were as follows: a 10 min initial denaturation step at 95 °C, followed by 40 three-step cycles, including a denaturation step (95 °C, 15 s), an annealing step (60 °C, 60 s), and an extension step (72 °C, 15 s). The experimental results were calculated using the 2^−ΔΔCt^ method.

### 2.10. Western Blot

Frozen synovial tissues were homogenized in cold RIPA lysis buffer containing 1 mM sodium orthovanadate and 1 mM PMSF on ice. The homogenates were centrifuged at 13,000× *g* for 10 min at 4 °C to remove debris. The total protein concentrations of the supernatants were quantified with BCA assay. Equal amounts of proteins (40 μg) were separated by 12% SDS–polyacrylamide gel electrophoresis and transferred to polyvinylidene difluoride (PVDF) membranes. The membranes were blocked with 5% (*w*/*v*) skimmed milk powder in Tris-buffered saline Tween 20 (TBST, 10 mM Tris, 150 mM NaCl, pH 7.4, 0.5% Tween 20) for 1.5 h, then were incubated with primary antibodies against mice, including GAPDH (1:10,000), TLR4 (1:2000), MyD88 (1:1000), p38 (1:1000), p-p38 (1:800), JNK (1:1000), p-JNK (1:1000), NF-κB p65 (1:1000), and p-p65 (1:500), at 4 °C overnight. After washing with TBST (3 × 10 min), the membranes were incubated with HRP-labeled goat anti-mouse or anti-rabbit secondary antibodies (1:1000) for 1.5 h at room temperature. The protein bands were developed with an ECL detection reagent according to the manufacturer’s instructions and were exposed to X-ray film. The intensity levels of antibody-reactive bands were analyzed with Image Lab 3.0 (Bio-Rad, Hercules, CA, USA).

### 2.11. Statistical Analysis

All data used were expressed as means ± standard deviation (SD). Statistical differences were analyzed by unpaired *t*-test with GraphPad Prism 5.0 software (GraphPad Software Inc., San Diego, CA, USA), where *p* < 0.05 was considered significantly different.

## 3. Results

### 3.1. Rosmanol and Carnosol Synergistically Alleviated RA Pathologies in CIA Mice

To examine the effects of rosmanol and carnosol on RA in vivo, a CIA DBA/1 mouse model was established. It was observed clearly that the development of arthritis was induced by BIIC on day 42, with the symptoms including swelling and redness (Figure 2b,c). Notably, the arthritis severity was alleviated in mice treated by rosmanol, carnosol, or the combination of both (Figure 2d–f). To compare the arthritis severity quantitatively, the arthritis index scores for each group were calculated on days 28, 35, and 42, respectively. The arthritis index score for the normal group was 0 on each day. On day 28, there were no significant differences among BIIC-induced groups. On day 35, compared to the model group, though the arthritis severity of the rosmanol-treated and carnosol-treated mice was attenuated slightly, for the combination-treated mice it was attenuated significantly (*p* < 0.05). On day 42, the arthritis severity was markedly attenuated in the rosmanol-treated and carnosol-treated group compared to the model group (*p* < 0.05), with arthritis index scores of 8.7, 8.5, and 10.0, respectively. In particular, the arthritis severity was further significantly attenuated in the combination-treated group with an arthritis index score of 6.8 (*p* < 0.05) (Figure 3). Taken together, these data clearly demonstrated that rosmanol and carnosol synergistically alleviated RA in CIA DBA/1 mice.

### 3.2. Rosmanol and Carnosol Synergistically Alleviated Synovitis in CIA Mice

To study the effects of rosmanol, carnosol, and their combination on synovitis, an H&E staining experiment was performed on mice synovial tissue sections. Representative results are shown in Figure 4. Compared to the normal group, severely proliferated synovial cells, hyperplastic fibrous tissues, and infiltrated inflammatory cells were observed in the model group (Figure 4a,b). These pathology changes were alleviated by daily gavage feeding with 40 mg/kg of rosmanol, carnosol, or a combination of both; the combination-treated mice in particular retained nearly normal architecture of synovial tissues (Figure 4c–e), suggesting that rosmanol and carnosol could synergistically alleviate synovitis in CIA DBA/1 mice.

### 3.3. Rosmanol and Carnosol Synergistically Decreased Pro-Inflammatory Cytokines in Serum

Persistent inflammation is maintained during RA pathogenesis [1]. To study the effects of rosmanol, carnosol, and their combination on the inflammation response of CIA mice, the serum levels of pro-inflammatory cytokines such as TNF-α, MCP-1, and IL-6 were quantified by ELISA on day 42 (Figure 5). These pro-inflammatory cytokines increased dramatically after stimulation by BIIC and decreased slightly in the rosmanol-treated group but decreased significantly in the carnosol-treated group (*p* < 0.01), indicating that rosmanol and carnosol exerted anti-inflammatory roles in CIA mice. Of note, the levels of TNF-α, MCP-1, and IL-6 further decreased in the combination-treated group (*p* < 0.05), indicating that rosmanol and carnosol synergistically inhibited the inflammation response in CIA DBA/1 mice.

### 3.4. Rosmanol and Carnosol Synergistically Inhibited the TLR4/NF-κB Pathway

TLR4/NF-κB pathway is an important regulator of pro-inflammatory gene expression, such as that of TNF-α, IL-6, IL-1β, and Cox-2. It plays a key role in the regulation of inflammation [18,19] and has been proven to be a therapeutic target for RA treatment [29]. To study the influences of rosmanol, carnosol, and their combination on the TLR4/NF-κB pathway, the transcription levels of TLR4, MyD88, and NF-κB p65 in synovial tissue were tested using real-time PCR on day 42. Compared to the normal group, the TLR4, MyD88, and NF-κB p65 mRNA levels of the model group were upregulated remarkably (*p* < 0.01). After treatment with rosmanol or carnosol, the transcription levels of TLR4, MyD88, and NF-κB p65 were downregulated to different degrees and the downregulation was strengthened when treated with rosmanol and carnosol together (*p* < 0.05) (Figure 6a–c). Similar results were obtained at the protein level, as examined by Western blot, whereby rosmanol and carnosol alone inhibited not only the translation of TLR4 and MyD88, but also the phosphorylation of p65. The inhibition was strengthened when mice were treated with their combination (*p* < 0.05) (Figure 6d–f). Taken together, these data demonstrate that rosmanol and carnosol synergistically inhibited the activation of the TLR4/NF-κB pathway.

### 3.5. Rosmanol and Carnosol Synergistically Inhibited MAPK Activation

MAPKs are downstream of TLR4 and play important roles in the pathogenesis of RA, as the abnormal activation of JNK and p38 almost participate in all aspects of RA-related pathologies [21,22,23,24]. To study the effects of rosmanol, carnosol, and their combination on the MAPKs, the expression levels of JNK, p-JNK, p38, and p-p38 in synovial tissue were examined by Western blot on day 42 (Figure 7). Compared to the normal group, though the expression levels of JNK and p38 were not influenced, the phosphorylation rates of JNK and p38 were upregulated remarkably in the model group (*p* < 0.01), indicating the activation of JNK and p38 MAPK pathways. The phosphorylation of JNK and p38 was inhibited by both rosmanol and carnosol alone (*p* < 0.05), while the inhibitory effect was enhanced when rosmanol and carnosol were used jointly (*p* < 0.05), indicating that rosmanol and carnosol synergistically inhibited the activation of JNK and p38 MAPK pathways.

## 4. Discussion

Rheumatoid arthritis is considered to be a multi-factorial autoimmune disease. Its pathogenesis involves epigenetic alterations, post-translational modifications, autophagy, T-cells, and other factors [30]. Natural products are abundant in quantity and diverse in structure, so they may target various biological processes; for example, curcumin and resveratrol are modulators of NF-κB [31,32], while Akone et al. reviewed the advances in natural products that could modulate DNA methylation and histone deacetylation [33], indicating that natural products are important sources for the discovery of drugs for the treatment of multi-factorial diseases such as RA.

*Callicarpa longissima* has been used as a Yao folk medicine to treat arthritis for a long time in China [6], although its anti-arthritic active moieties have not been clarified so far. In this study, we reported on anti-RA constituents and their active mechanism of this ethnomedicine for the first time. Rosmanol and carnosol, two major phenolic diterpenoids from the branches and leaves of *C. longissima*, were found to alleviate swelling, redness, and synovitis in CIA DBA/1 mice (Figure 2, Figure 3 and Figure 4), demonstrating that they were the constituents responsible for the anti-RA effect of *C. longissima*. The safety of rosmanol and carnosol has been proven in previous studies [13,14]. In this study, though rosmanol and carnosol exhibited a slight ability to alleviate body weight loss in CIA mice, their combination significantly alleviated body weight loss after day 35 (Appendix A), indicating the safety of rosmanol, carnosol, and their combination. Therefore, they have the potential to be developed into novel, safe natural agents for the treatment of RA.

A synergistic effect is an interaction of two or more ingredients that results in a greater effect than the total of their separate effects. Drug combinations with synergistic effects can often alleviate drug resistance and improve the therapeutic efficacy [34], so it is highly significant to continue to research such combinations. Rosmanol and carnosol synergistically inhibited paw swelling (Appendix A); inspired by this discovery, the anti-RA effect of their combination was also assessed in this study. Compared to rosmanol or carnosol alone, their combination alleviated RA symptoms such as swelling, redness, and synovitis and reduced the arthritis index score more significantly (Figure 2, Figure 3 and Figure 4). In addition, on day 35, the RA-related pathologies in the combination-treated mice were significantly alleviated, although the inhibitory effect was not obvious in the rosmanol-treated or carnosol-treated mice (Figure 3). These data powerfully demonstrated that rosmanol and carnosol synergistically alleviated RA in CIA DBA/1 mice.

It is well-known that the inflammation response, which is involved in many pro-inflammatory mediators, takes part in every phase of RA pathogenesis [1]. Compared to rosmanol-treated and carnosol-treated groups, the serum levels of TNF-α, IL-6, and MCP-1 in the combination-treated mice were all significantly reduced (Figure 5), indicating that rosmanol and carnosol could synergistically alleviate RA via inhibiting the inflammation response. This effect was similar to those of their respective analogs rosmarinic acid and carnosic acid, whose anti-RA activity also via inflammation suppression [35,36].

Earlier studies have demonstrated that the TLR4/NF-κB pathway is a plausible therapeutic target for RA treatment [29,37]. The combination of rosmanol and carnosol suppressed both the expression levels of TLR4 and MyD88 and the phosphorylation rates of NF-κB p65 more significantly than either compound alone (Figure 6), indicating that they synergistically inhibited the activation of the TLR4/NF-κB pathway in CIA DBA/1 mice. The combination of rosmanol and carnosol also blocked the phosphorylation rates of both JNK and p38 more powerfully than either single compound did (Figure 7), indicating that they synergistically inhibited the activation of both JNK and p38 MAPK pathways. Rosmanol and carnosol could possibly inhibit angiogenesis, cartilage degradation, and bone destruction, as the activation of JNK and p38 involves these RA-related pathologies [21,22,23,24], although this speculation requires further validation.

In conclusion, the data presented in this study clearly demonstrate that rosmanol and carnosol synergistically alleviated RA via inhibiting inflammation through regulating the TLR4/NF-κB/MAPK pathway (Figure 8), and they have the potential to be developed into novel, safe natural combinations for the treatment of RA. The results also proved that *Callicarpa longissima* can be used as a source of natural anti-inflammatory drugs.

## Figures and Tables

**Figure 1 molecules-27-00078-f001:**
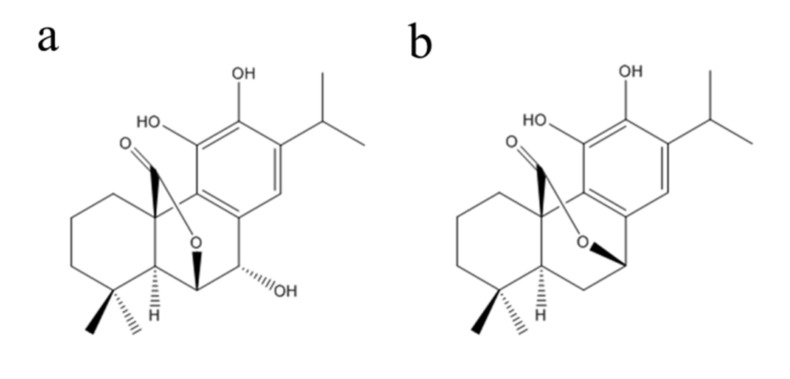
Chemical structures of rosmanol (**a**) and carnosol (**b**).

**Figure 2 molecules-27-00078-f002:**
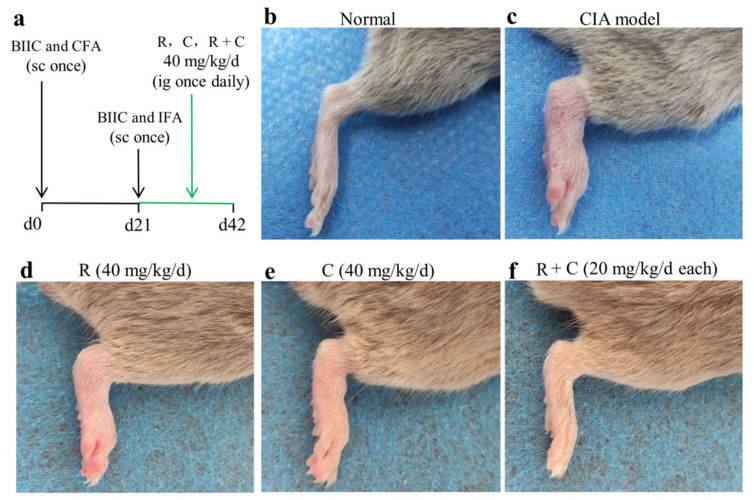
Rosmanol, carnosol, and their combination alleviated RA severity in CIA mice. Schematic of establishment of CIA DBA/1 mouse model and the treatments of rosmanol (R), carnosol (C), and their combination (R + C) in the CIA mice (**a**). Representative pictures from the right hind limbs of the normal group (**b**), model group (**c**), rosmanol-treated group (**d**), carnosol-treated group (**e**), and combination-treated group (**f**) on day 42.

**Figure 3 molecules-27-00078-f003:**
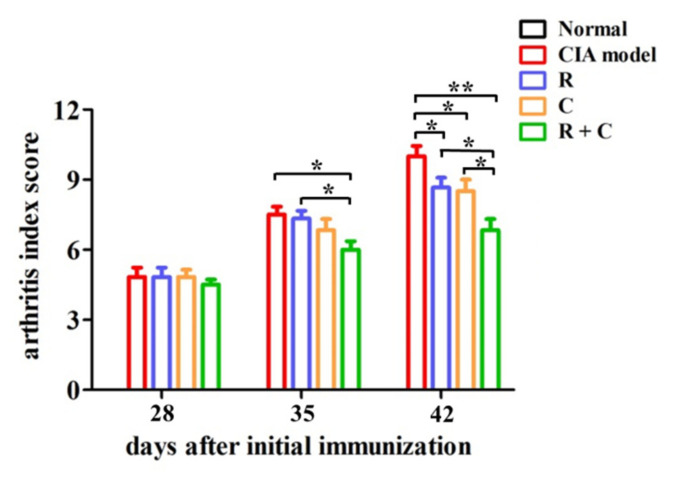
Effects of rosmanol (R), carnosol (C), and their combination (R + C) on arthritis index scores in CIA mice. Arthritis index scores were the cumulative scores for all four paws, which represented the overall disease severity. The arthritis index score for each group is presented as the mean ± SD (*n* = 6). The significance of differences was analyzed by unpaired *t*-test. Note: * *p* < 0.05; ** *p* < 0.01.

**Figure 4 molecules-27-00078-f004:**
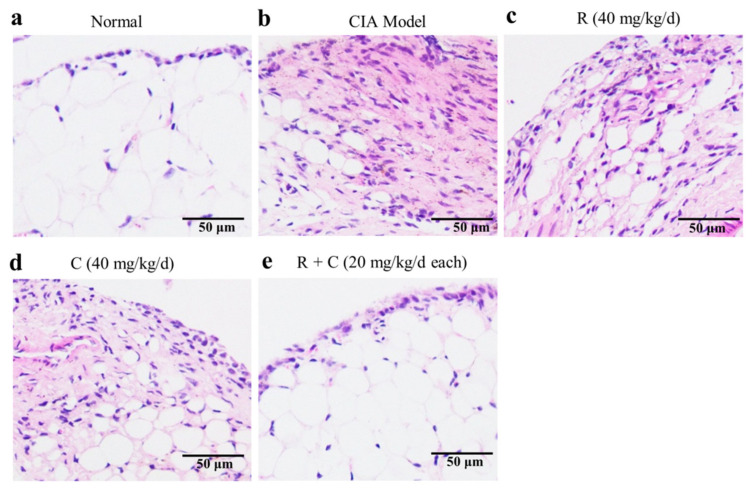
Rosmanol (R), carnosol (C), and their combination (R + C) alleviated synovitis in CIA mice. Representative H&E staining results from synovial tissue samples of the normal group (**a**), model group (**b**), rosmanol-treated group (**c**), carnosol-treated group (**d**), and combination-treated group (**e**) mice are shown (200×, *n* = 3).

**Figure 5 molecules-27-00078-f005:**
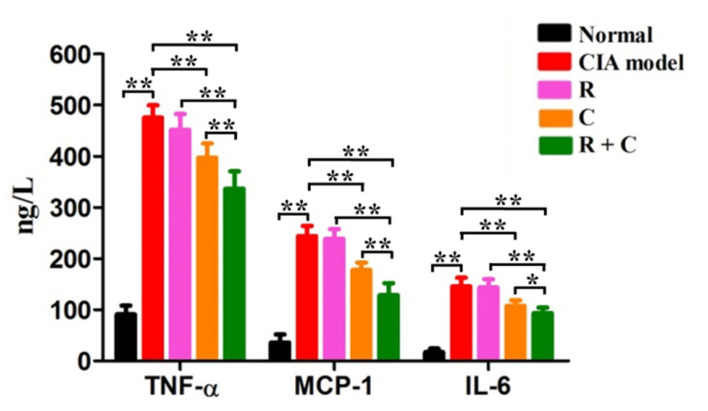
The effects of rosmanol (R), carnosol (C), and their combination (R + C) on the production of TNF-α, MCP-1, and IL-6. The serum levels of TNF-α, MCP-1, and IL-6 of each group were determined by ELISA on day 42. Data are presented as the means ± SD (*n* = 6). The significance of differences was analyzed by unpaired *t*-test. Note: * *p* < 0.05; ** *p* < 0.01.

**Figure 6 molecules-27-00078-f006:**
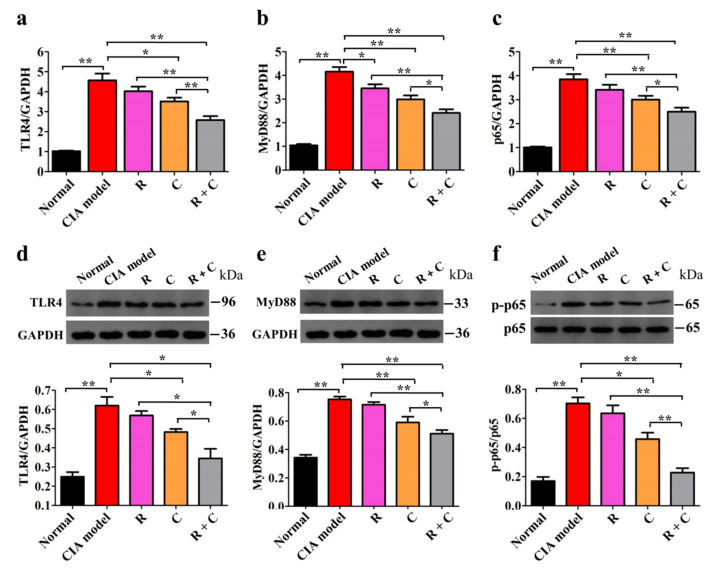
Rosmanol (R), carnosol (C), and their combination (R + C) inhibited the TLR4/NF-κB pathway. The relative transcription levels of TLR4 (**a**), MyD88 (**b**), and NF-κB p65 (**c**) in synovial tissue samples of each group were quantified by real-time PCR on day 42. The representative Western blot analysis results and protein relative expression levels for TLR4 (**d**), MyD88 (**e**), and NF-κB p-p65 (**f**) in synovial tissue samples of each group on day 42 are shown. GAPDH was used as an internal reference to quantify the expression levels of TLR4 and MyD88. The phosphorylation of p65 was expressed as p-p65/p65. Data are presented as the mean ± SD (*n* = 3). The significance of differences was analyzed by unpaired *t*-test. Note: *, *p* < 0.05; **, *p* < 0.01.

**Figure 7 molecules-27-00078-f007:**
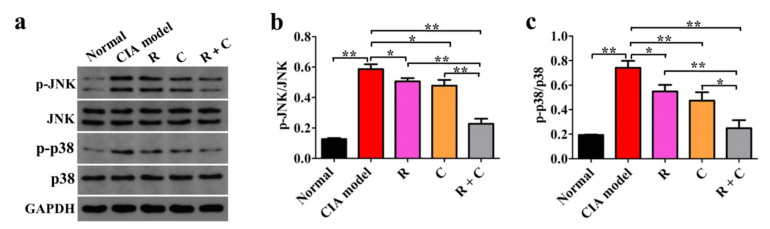
Rosmanol (R), carnosol (C), and their combination (R + C) inhibited the JNK and p38 MAPK pathways. The protein expression levels of JNK, p-JNK, p38, and p-p38 in synovial tissue samples of each group were examined by Western blot on day 42 (**a**) and the relative phosphorylation rates of JNK (**b**) and p38 (**c**) were calculated. GAPDH was used as the internal reference to quantify the expression levels of JNK, p-JNK, p38, and p-p38. The phosphorylation rates of JNK and p38 were expressed as p-JNK/JNK and p-p38/p38, respectively. Data are presented as the means ± SD (*n* = 3). The significance of differences was analyzed by unpaired *t*-test. Note: * *p* < 0.05; ** *p* < 0.01.

**Figure 8 molecules-27-00078-f008:**
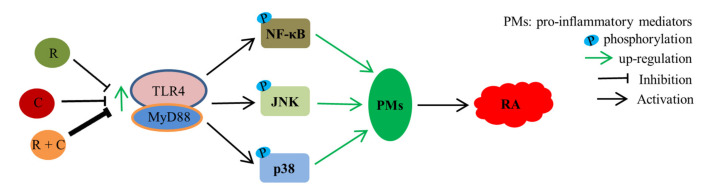
The signaling pathways that rosmanol (R), carnosol (C), and their combination (R + C) target in RA. The intensity of inhibition is proportional to the line width of the inhibition symbol.

## Data Availability

Data are provided in the manuscript and Appendix A.

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
