# Peer review of "Rosmanol and Carnosol Synergistically Alleviate Rheumatoid Arthritis through Inhibiting TLR4/NF-κB/MAPK Pathway"

_molecules, 2021, doi:10.3390/molecules27010078_

Round 1
Reviewer 1 Report
This manuscript is suitable for publication in the journal with some revisions.
Introduction
Line 48-50: Please provide background information/evidence that the two isolated phenolic terpenoids are highly abundant in the medicinal plant under study.
Line 52-54: Please offer more information about the two diterpenoids' safety since they may exhibit a concentration-dependent effect associated with toxicity in some cases.
Methods
The authors should describe the analytical conditions used for the structure elucidation of the isolated compounds or reference the method applied. In addition, instead of only writing the results, authors should also present the spectrum(s). I understand this was not the purpose of this study, but it is more appropriate.
Results and discussion
The toxicity assessment of the combination-tested is necessary to infer the treatment safety. To have the confidence of the toxicity of the treatment can largely improve the liability of the result.
Author Response
Introduction
Point 1: Line 48-50: Please provide background information/evidence that the two isolated phenolic terpenoids are highly abundant in the medicinal plant under study.
Response 1: Thanks for your reminder. The contents determination of rosmanol and carnosol in Callicarpa longissima was described in supplementary materials of the revised manuscript.
Point 2: Line 52-54: Please offer more information about the two diterpenoids' safety since they may exhibit a concentration-dependent effect associated with toxicity in some cases.
Response 2: Thanks for your reminder. The references about acute toxicity of rosmanol and carnosol were cited.
Methods
Point 3: The authors should describe the analytical conditions used for the structure elucidation of the isolated compounds or reference the method applied. In addition, instead of only writing the results, authors should also present the spectrum(s). I understand this was not the purpose of this study, but it is more appropriate.
Response 3: Thanks for your suggestion. The HPLC, MS and NMR spectra of rosmanol and carnosol were presented in supplementary materials of the revised manuscript.
Results and discussion
Point 4: The toxicity assessment of the combination-tested is necessary to infer the treatment safety. To have the confidence of the toxicity of the treatment can largely improve the liability of the result.
Response 4: The safety of rosmanol and carnosol has been proven by previous studies (Abdelhalim et al. J. Pharm. Pharm. Sci. 2015, 18: 448–459; Khan et al. Evid. Based Complement. Alternat. Med. 2016, 2016: 1215393). From the view of chemistry, rosmanol and carnosol are unlikely to react. So when they are used together, it’s unlikely to produce toxic substances. And in this study, their combination could significantly alleviate the body weight loss after day 35 in CIA mice (Table S1), indirectly demonstrating the safety of the combination.
Reviewer 2 Report
The manuscript,” Rosmanol and carnosol synergistically alleviate rheumatoid arthritis through inhibiting TLR4/NF-κB/MAPKs pathway”, is written well. The concept, methodology, results and conclusion are well written. Few mistakes been observed like:
Line 11: anti-arthritic active moieties
13: medicinal plant
28: that is characterized by presence of auto…
68: paw edema
74: we assume rosmanol.
Similar should be checked and rectified.
English language and grammar should be considered.
Author Response
Point 1: The manuscript,” Rosmanol and carnosol synergistically alleviate rheumatoid arthritis through inhibiting TLR4/NF-κB/MAPKs pathway”, is written well. The concept, methodology, results and conclusion are well written. Few mistakes been observed like:
Line 11: anti-arthritic active moieties
13: medicinal plant
28: that is characterized by presence of auto…
68: paw edema
74: we assume rosmanol.
Similar should be checked and rectified. English language and grammar should be considered.
Response 1: Thanks for your well-intentioned reminder. We have carefully checked English language for spelling, phrase, punctuation and grammar repeatedly. And the revisions to the manuscript have been marked up using the “Track Changes” function.
Reviewer 3 Report
Title: Rosmanol and carnosol synergistically alleviate rheumatoid arthritis through inhibiting TLR4/NF-κB/MAPKs pathway
Authors: Lian-Chun Li, Zheng-Hong Pan, De-Sheng Ning, Yu-Xia Fu
Summary:
The authors showed that rosmanol and carnosol synergistically alleviate RA by inhibiting inflammation via regulating the TLR4/NF-κB/MAPKs pathway and have the potential to be developed into a new safe natural combination for the treatment of RA.
Comments:
Indeed, it is interesting to investigate the potential effect of polyphenols on RA by inhibiting inflammation via regulating the TLR4/NF-κB/MAPKs pathway. The figures are very nicely presented and show impressive results.
Important points are listed below:
1: Page 1, line 37: "...anti-RA medicinal plants" I don't like the wording as these plants target specific signaling pathways/mechanisms much more than the disease itself, i.e. if they are anti-inflammatory, for example, they can be used for all diseases where inflammation plays a role
Suggestion: "...more and more anti-inflammatory plants traditionally used to treat RA have been considered."
2: Page 1, line 39: I miss in this nice work (Introduction and discussion) a section about specific natural anti-inflammatory, inhibitors and modulators of NF-kB, like curcumin or resveratrol in chronic diseases and RA. This would be very good if the authors discuss the natural modulation (nuclear translocation, phosphorylation, histone modification, DNA binding) an extra section.
Please add additional reference:
doi.org/10.3390/cells10113017.
doi: 10.3390/ijms22147645.
doi: 10.1016/j.bcp.2008.05.029.
3: Page 1, line 39-40: Here I miss a transition: what connects "Tripterygium Wilfordii" and "Callicarpa longissima" in the next line? Here a jump is made from one plant to another without taking the reader along.
4: Page 2, line 54: Are the authors sure that the substances are unrestricted safe without having tested it? There are toxic concentrations even with natural substances.
5: Page 2, line 56-71: make it clearer that these are earlier studies.
6: Page 2 line 57 + 58: again, I don't like the wording; there are no studies (to my knowledge) that specifically test RA and carnosol/rosmanol, but there are studies that examine the effect that also plays a role in RA/inflammation, e.g. Sanchez C, Horcajada MN, Membrez Scalfo F, Ameye L, Offord E, and Henrotin Y, Carnosol inhibits pro-inflammatory and catabolic mediators of cartilage degradation in human osteoarthritic chondrocytes and mediates cross-talk between subchondral bone osteoblasts and chondrocytes.
7: Page. 2, line 70: Wrong spelling of anti-inflammatory.
8: I miss a link in the introduction to the Pathways mentioned in the heading.
9: Page 3, line 112: remove period after biontech.
10: 2.5: How does the selection of concentrations of natural products come about? Wouldn't one have to do a concentration dependent experiment first to make a selection?
11: Page 4, Figure 2c, Figure 4b, Figure 6d, Figure 7a: here I would write "CIA model" instead of just "model" (very non-specific).
12: Page 6, Figure 3: here I miss an axis label for the x-axis or "tag" should be written out differently
13: Page 7, Figure 4 d + e: scales not visible/cut off.
14: The headings of the results (3.1, 3.2, 3.3) are very unspecific and should outline the respective result more precisely Synovitis (3.2.) is e.g. also an inflammation (see chapter 3.3. "Inflammation")
15: Figure 1 in the introduction seems unusual to me; I would rather show the structural formulas for chapter 2 (2.2.), since this is about rosmanol and carnosol in detail
16: Page 6, Fig. 3: Was the arthritis index value in the normal mice 0 on each day? It is not shown here for comparison.
17: Page 8, line 258: Why is the TLR4/NF-kB pathway a therapeutic target for RA? (with brief rationale)
18: Page 9, line 278: Briefly explain why MAPKs play an important role in RA.
19: Figurs 6 & 7: Mention the function of p65, p38 and GAPDH in WB.
20: Page 9, lines 296 & 300: adjust formatting.
21: I would appreciate a schematic overview of the signaling pathways and target components in RA as this would address the title topic.
22: Page 10, line 343: Do the authors suggest natural products as the sole treatment or as an adjunct to classical drugs? What are the next steps planned?
More:
23: Language and grammar should be reviewed. Singular versus plural, spelling. All text needs to be revised by a native English speaker.
partly missing words in sentences, e.g. page 1 line 28 "..chronic autoimmune disease that characterized..." (an "is" is missing); use of wrong adjective forms, e.g. page 2 line 50 "...and carnosol are riched in this..." (it should be "rich" or "enriched", same on p. 9 line 300);
24: A list of abbreviations would be helpful.
Author Response
Point 1: Page 1, line 37: "...anti-RA medicinal plants" I don't like the wording as these plants target specific signaling pathways/mechanisms much more than the disease itself, i.e. if they are anti-inflammatory, for example, they can be used for all diseases where inflammation plays a role.
Suggestion: "...more and more anti-inflammatory plants traditionally used to treat RA have been considered."
Response 1: That’s a good suggestion. We have adopted it in the revised manuscript.
Point 2: Page 1, line 39: I miss in this nice work (Introduction and discussion) a section about specific natural anti-inflammatory, inhibitors and modulators of NF-kB, like curcumin or resveratrol in chronic diseases and RA. This would be very good if the authors discuss the natural modulation (nuclear translocation, phosphorylation, histone modification, DNA binding) an extra section.
Please add additional reference:
doi.org/10.3390/cells10113017.
doi: 10.3390/ijms22147645.
doi: 10.1016/j.bcp.2008.05.029.
Response 2: Thanks for your suggestion. A section about natural modulator was added in discussion.
Point 3: Page 1, line 39-40: Here I miss a transition: what connects "Tripterygium Wilfordii" and "Callicarpa longissima" in the next line? Here a jump is made from one plant to another without taking the reader along.
Response 3: Sorry for our mistake. A transitional sentence was added in the revised manuscript.
Point 4: Page 2, line 54: Are the authors sure that the substances are unrestricted safe without having tested it? There are toxic concentrations even with natural substances.
Response 4: We agree that there are toxic concentrations even with natural substances. We didn’t test the safety of rosmanol and carnosol because earlier studies had showed that rosmanol and rosmanol and carnosol in the dose range of 50-200 mg/kg/d did not exhibit any signs of acute toxicity in male Swiss mice (Abdelhalim et al. J. Pharm. Pharm. Sci. 2015, 18: 448–459; Khan et al. Evid. Based Complement. Alternat. Med. 2016, 2016: 1215393). And we cited the results of these studies in the revised manuscript.
Point 5: Page 2, line 56-71: make it clearer that these are earlier studies.
Response 5: Some slight modifications have been made in the revised manuscript.
Point 6: Page 2 line 57 + 58: again, I don't like the wording; there are no studies (to my knowledge) that specifically test RA and carnosol/rosmanol, but there are studies that examine the effect that also plays a role in RA/inflammation, e.g. Sanchez C, Horcajada MN, Membrez Scalfo F, Ameye L, Offord E, and Henrotin Y, Carnosol inhibits pro-inflammatory and catabolic mediators of cartilage degradation in human osteoarthritic chondrocytes and mediates cross-talk between subchondral bone osteoblasts and chondrocytes.
Response 6: We have deleted the sentence” but no study related to the anti-RA effect of either rosmanol or carnosol has been reported yet”.
Point 7: Page. 2, line 70: Wrong spelling of anti-inflammatory.
Response 7: Sorry for our carelessness, we have made it right.
Point 8: I miss a link in the introduction to the Pathways mentioned in the heading.
Response 8: Thanks for your suggestion. A section about TLR4/ NF-κB/MAPKs pathway was briefly described in the introduction part of the revised manuscript.
Point 9: Page 3, line 112: remove period after biontech.
Response 9: We have corrected it in the revised manuscript.
Point 10: 2.5: How does the selection of concentrations of natural products come about? Wouldn't one have to do a concentration dependent experiment first to make a selection?
Response 10: In fact, in our previous study with carrageen-induced paw edema mice model, we designed several concentrations for rosmanol, carnosol and their combination. And there were various ratios of rosmanol to carnosol in the combination. The results showed when the doses were no less than 40 mg/kg/d, and the ratio of rosmanol to carnosol was 1:1, the combination had much powerful anti-inflammatory activity than either of the single compound. Due to some reasons, the data is still in the process of preparing for publication. To save compounds and experimental cost, we chose 40 mg/kg/d as the dose of rosmanol, carnosol and their combination in this study.
Point 11: Page 4, Figure 2c, Figure 4b, Figure 6d, Figure 7a: here I would write "CIA model" instead of just "model" (very non-specific).
Response 11: Thank for your suggestion. We have adopted it in the revised manuscript.
Point 12: Page 6, Figure 3: here I miss an axis label for the x-axis or "tag" should be written out differently
Response 12: A tag for the x-axis has been added.
Point 13: Page 7, Figure 4 d + e: scales not visible/cut off.
Response 13: Maybe the scales are too slender. We have made them bold.
Point 14: The headings of the results (3.1, 3.2, 3.3) are very unspecific and should outline the respective result more precisely Synovitis (3.2.) is e.g. also an inflammation (see chapter 3.3. "Inflammation")
Response 14: Thanks for your reminder. To make the headings of the results specific, we have made some revision to them.
Point 15: Figure 1 in the introduction seems unusual to me; I would rather show the structural formulas for chapter 2 (2.2.), since this is about rosmanol and carnosol in detail
Response 15: Thank for your suggestion. We have adopted it in the revised manuscript.
Point 16: Page 6, Fig. 3: Was the arthritis index value in the normal mice 0 on each day? It is not shown here for comparison.
Response 16: Yes, the arthritis index score of normal mice was 0 on each day, and it was shown for comparison in the revised manuscript.
Point 17: Page 8, line 258: Why is the TLR4/NF-kB pathway a therapeutic target for RA? (with brief rationale)
Response 17: TLR4/NF-κB pathway is an important regulator of pro-inflammatory gene expression such as TNF-α, IL-6, IL-1β and Cox-2, and plays key role for the regulation of inflammation. So, it’s a therapeutic target for RA.
Point 18: Page 9, line 278: Briefly explain why MAPKs play an important role in RA.
Response 18: MAPKs play an important role in RA, as their activation almost involves in all aspects of RA-related pathologies.
Point 19: Figurs 6 & 7: Mention the function of p65, p38 and GAPDH in WB.
Response 19: Thanks for your reminder. The functions of p65, JNK, p38 and GAPDH have been described in legends of Fig. 6 & 7.
Point 20: Page 9, lines 296 & 300: adjust formatting.
Response 20: The format has been adjusted.
Point 21: I would appreciate a schematic overview of the signaling pathways and target components in RA as this would address the title topic.
Response 21: That’s a good suggestion, and a schematic was presented in the revised manuscript.
Point 22: Page 10, line 343: Do the authors suggest natural products as the sole treatment or as an adjunct to classical drugs? What are the next steps planned?
Response 22: Because the presented data in the manuscript was collected from animal experiments, we are not sure whether rosmanol and carnosol may be used as the sole treatment. Therefore, we are trying to obtain evaluation and advice from experts who work on drug research and development.
More:
Point 23: Language and grammar should be reviewed. Singular versus plural, spelling. All text needs to be revised by a native English speaker.
partly missing words in sentences, e.g. page 1 line 28 "..chronic autoimmune disease that characterized..." (an "is" is missing); use of wrong adjective forms, e.g. page 2 line 50 "...and carnosol are riched in this..." (it should be "rich" or "enriched", same on p. 9 line 300);
Response 23: Thanks for your kind reminder. We have carefully checked English language for spelling, punctuation, phrase and grammar repeatedly. And the revisions to the manuscript have been marked up using the “Track Changes” function.
Point 24: A list of abbreviations would be helpful.
Response 24: That is a good suggestion. And a list of abbreviations was placed after the keywords in the revised manuscript.
Reviewer 4 Report
I have carefully read the manuscript and have no further comments or requests regarding the scientific and experimental parts. authors should check English language for spelling and grammar errors (articles).
For example: line 28, that characterized by... it should be: that is characterized by...
Author Response
Point 1: I have carefully read the manuscript and have no further comments or requests regarding the scientific and experimental parts. Authors should check English language for spelling and grammar errors (articles).For example: line 28, that characterized by... it should be: that is characterized by...
Response 1: Thanks for your kind reminder. We have carefully checked English language for spelling, punctuation, phrase and grammar repeatedly. And the revisions to the manuscript have been marked up using the “Track Changes” function.
Round 2
Reviewer 3 Report
The authors have satisfactorily addressed the concerns raised in the original version. The revised version is significantly improved. No further concerns.